# Research

evolution

adaptive radiation, eusociality, phylogeny, social insects, social evolution

**Author for correspondence:**
Nobuaki Mizumoto
e-mail: nobuaki.mzmt@gmail.com

# The evolution of body size in termites

Nobuaki Mizumoto and Thomas Bourguignon

Okinawa Institute of Science and Technology Graduate University, 1919-1 Tancha, Onna-son, Okinawa 904-0495, Japan

 NM, 0000-0002-6731-8684; TB, 0000-0002-4035-8977

Termites are social cockroaches. Because non-termite cockroaches are larger than basal termite lineages, which themselves include large termite species, it has been proposed that termites experienced a unidirectional body size reduction since they evolved eusociality. However, the validity of this hypothesis remains untested in a phylogenetic framework. Here, we reconstructed termite body size evolution using head width measurements of 1638 modern and fossil termite species. We found that the unidirectional body size reduction model was only supported by analyses excluding fossil species. Analyses including fossil species suggested that body size diversified along with speciation events and estimated that the size of the common ancestor of modern termites was comparable to that of modern species. Our analyses further revealed that body size variability among species, but not body size reduction, is associated with features attributed to advanced termite societies. Our results suggest that miniaturization took place at the origin of termites, while subsequent complexification of termite societies did not lead to further body size reduction.

## 1. Introduction

Body size diversification is an indicator of ecological diversification [1–3]. Diversification occurs when new resources or niches become available [4,5], often leading to the evolution of new phenotypes (i.e. key innovations [6,7]). The evolution of eusociality is a major evolutionary transition [8], which potentially leads to phenotypic diversification [9,10], including body size diversification. Social insects, especially ants and termites, are among the most successful and abundant terrestrial animals [11]. Their colonies typically contain many individuals belonging to distinct specialized phenotypic castes, which vary in size in a species-specific manner. However, the factors responsible for body size variation among species, and the role of social evolution, remain unclear. This problem can be addressed by analyses of body size measurements in a comparative phylogenetic framework.

Termites are a group of social insects that share a common ancestor with the wood roach *Cryptocercus*, from which they diverged some 170 Ma [12–14]. The genus *Cryptocercus* contains only 12 of the 4600 described non-termite cockroach species [15], while termites include over 3000 described modern species [16], considerably varying in body size (figure 1). The unidirectional reduction in body size hypothesis is believed to be a general evolutionary trend in termites [17]. Small body size enables more individuals to inhabit small pieces of wood, perhaps allowing larger and more complex societies to evolve [18]. Small body size also allows for saving nutrients, especially nitrogen, which is limiting for wood-boring insects [17]. The unidirectional body size reduction hypothesis is supported by the observations that all species of *Cryptocercus* are much larger than species of termites (figure 1), and modern representatives of early-diverging termite lineages are generally large [17,19]. However, this idea has never been adequately tested and relies on comparisons among termite families and cockroaches. In addition, fossil species have not been taken into consideration, despite the existence of key fossil species with small body size,

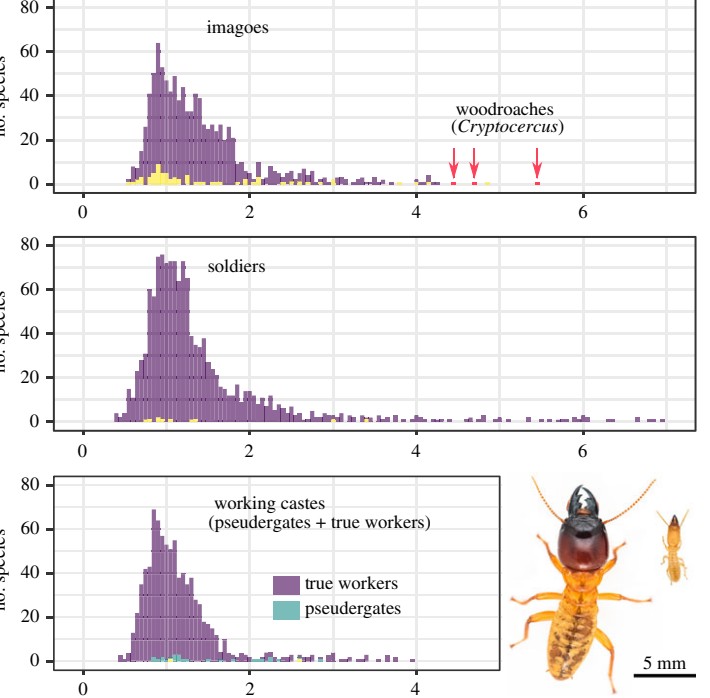

**Figure 1.** Distribution of head width across termite species. We used the data of the largest morph when within-caste polymorphism was present. The photos show the soldiers of a large species (*Hodotermopsis sjostedti*) and a small species (*Reticulitermes okinawanus*) (photos: Ales Bucek). (Online version in colour.)

such as *Melqartitermes*, one of the oldest know termite fossils [20], and *Nanotermes*, the smallest termite species ever known to have existed [21].

Another factor that has possibly affected body size evolution in termites is sociality. While all termites are eusocial, the level of social complexity varies among species and can be roughly approximated by nesting strategies and developmental pathways (figure 3a). Social complexity is presumably the lowest in one-piece nesting termites that nest in the piece of wood on which they feed, followed by multiple-piece nesting termites that feed on multiple wood items connected by networks of galleries, and is the highest in separate-piece nesting termites that build large nest structures separated from their food sources [22,23] (figure 3a). The construction of large nests, separated from food sources, is undoubtedly a derived trait in termites [23,24], enabling colonies to prosper over long periods. Developmental pathways are also variable among termite species, with two distinct types: the linear developmental pathway and the forked developmental pathway [25,26] (figure 3a). Species with a linear developmental pathway are often considered socially primitive as they lack a true worker caste. In these species, colony tasks are performed by immatures, called 'pseudergates', which retain the potential to develop into alate imagoes. By contrast, species with a forked developmental pathway possess a caste of 'true workers' that irreversibly deviates from the imaginal developmental line at an early developmental stage and cannot moult into alate imagoes, although they are still able to reproduce in most lower termite species and some higher termite species [27]. Owing to this additional caste, species with true workers have increased phenotypic and behavioural plasticity [28], potentially allowing for the evolution of more complex social systems. Whether a separate-piece nesting strategy and the presence of a true worker caste are linked to body size evolution remains unclear.

In this study, we reconstructed termite body size evolution using head width data collected from 153 papers (electronic supplementary material, data S1). We used head width as a proxy for body size because it has been consistently measured since the inception of termite taxonomy [17,29]. In addition, head width is strongly correlated with body mass among different termite species (electronic supplementary material, text and figure S2). We fit seven evolutionary models on two phylogenetic trees, reconstructed with and without fossil species, to identify the most plausible scenario of body size evolution in termites. More precisely, we tested whether the unidirectional body size reduction model explains body size evolution across the termite phylogeny, or at least across the lower termite phylogeny, excluding higher termites, which have the largest body size diversity among termite families. We also examined whether characteristics traditionally attributed to complex termite societies, including separate-piece nesting strategy, the presence of a true worker caste and large colony size, are linked to body size evolution. Furthermore, we investigated how termite social evolution has shaped body size variation among castes, including body size variation among alate imagoes, soldiers and working castes (pseudergates or true workers).

## 2. Methods

### (a) Data collection

We collected termite head width data from the literature, mainly from taxonomic papers cited in the Termite Database [30]. We obtained one representative value for each caste (workers/pseudergates, soldiers and alate imagoes) of every species. We used species mean head width values, mid-range values (calculated from the minimum and maximum values for the species for which only ranges were reported), or head width of the

holotype, in this order of priority. Thus, we did not exclude species with a small number of head width measurements. When measurements were available from multiple sources, we used all measurements to calculate the mean value among sources. For species displaying polymorphism within the worker or soldier castes, we used the measurements from the largest subcaste (i.e. major workers and major soldiers) to keep consistency across species. In total, 12.39% (554 of 4471 data points) of the measurements were derived from polymorphic species. Analyses performed using measurements from the smallest subcaste yielded similar results leaving our conclusions unchanged. Workers and soldiers usually lack eyes, while eyes may be included or excluded from head width measurements of imagoes. We used head width data of alate imagoes including eyes, and excluded data explicitly taken without eyes. We assumed that eyes were included in head width measurements when the authors made no mention of eyes because 'maximum width of head with eyes' has been the recommended measurement in termite taxonomy [29], and most studies include eyes in the measurements. The whole dataset and the source of measurements are available in electronic supplementary material, data S1-2.

We classified termites based on their nesting habitats, the presence of true workers and colony size. We recognized three categories of termite nesting habitats, as described by Abe [22]: one-piece, multiple-piece and separate-piece nesters (figure 3a). One-piece nesters include *Zootermopsis*, all genera of Stolotermitidae, Stylotermitidae, Serritermitidae, almost all species of Kalotermitidae, *Prorhinotermes* and *Termitogeton*; multiple-piece nesters include *Mastotermes*, *Hodotermopsis*, *Paraneotermes* and most species of Rhinotermitidae; and the separate-piece nesters include all Hodotermitidae and Termitidae. Similarly, we classified termites into two categories based on the presence of true workers (figure 3a); *Mastotermes*, Hodotermitidae, Rhinotermitinae, *Reticulitermes*, *Heterotermes*, *Coptotermes* and all Termitidae have true workers, while other taxa rely on pseudergates for colony tasks. We obtained maximum colony size estimates from one previous study [31]. Note that the methods used to infer colony size varied among studies and are prone to errors.

## (b) Phylogeny

We used MRBAYES v. 3.2.7 [32] to reconstruct a time-calibrated phylogenetic tree, using a relaxed clock model that combined molecular data of extant termite species and morphological characters of extant and fossil termite species. Our phylogenetic tree was composed of 183 taxa, including 138 modern termite species, each belonging to a distinct genus, 39 fossil termite species, and six outgroups, including *Cryptocercus*, four other cockroaches, and one mantis. The molecular data included 139 (133 termites + six outgroups) mitochondrial genomes (references are in electronic supplementary material, Data S3). All mitochondrial genomes were annotated using the MITOS webserver [33] with the invertebrate mitochondrial genetic code and default parameters. The two ribosomal RNA genes, 22 transfer RNA genes, and 13 protein-coding genes were aligned independently with MAFFT v7.300b using the options '–maxiterate 1000 –globalpair' for maximum accuracy [34]. Ribosomal RNA genes and transfer RNA genes were aligned as DNA sequences. Protein-coding genes were aligned as protein sequences and were back-translated to DNA sequences using pal2nal v14 [35]. We used the dataset in [36] for the morphological data, which included 111 morphological characters scored across 82 taxa with five outgroups. We reduced the dataset to 79 taxa, including four outgroups. Modern taxa in this dataset were associated with one mitochondrial genome derived from the same species or another congeneric species when available. Because all modern species used in this study belonged to distinct genera, the morphological and molecular data always derived from specimens that were more closely related

to each other than to any other taxa used for phylogenetic inferences. Fossil taxa were coded as '?' for molecular characters, and extant taxa without morphological data were coded as '?' for morphological characters. Our final dataset included 183 taxa, 145 of which were living taxa and 38 were fossil taxa. Both molecular and morphological data were available for 35 taxa, while 104 taxa and 44 taxa were exclusively represented by molecular data and morphological data, respectively. The molecular dataset was partitioned into four subsets: one combining the 12S and 16S rRNA genes, one combining the 22 tRNA genes, one for the first codon sites of protein-coding genes, and one for the second codon sites of protein-coding genes. The third codon position sites were excluded from the phylogenetic reconstruction because their substitution rate is too high to infer old divergences reliably. We used a GTR model with gamma-distributed rate variation across sites. The morphological data were assigned a MK $+ \Gamma$ model, with coding set to 'variable' to account for acquisition bias [1,37]. To assist the analyses, we applied a series of topological constraints based on the topology reported in previous studies [13,14,36]. We used a consensus tree for the formal analysis. Note that we performed the analyses described below using a phylogenetic tree reconstructed without topological constraints and found similar results. We also performed the analysis using five trees sampled randomly along the MRBAYES MCMC chain to account for the uncertainties of tree topology, especially for the placement of fossil species that was based on a limited set of morphological characters. The results were highly congruent among analyses (data not shown). The nexus file, including the final alignment and the MRBAYES block, and the reconstructed phylogenetic tree are available as supplementary material (electronic supplementary material, Data S4 and S5).

## (c) Modelling of body size evolution

We combined our head width dataset with our phylogenetic tree to investigate various scenarios of termite body size evolution. We used imago head width as the representative body size measurements for each species because imago is the adult stage and the most common caste found in the fossil record. Taxa lacking head width data and taxa missing in our phylogenetic tree were excluded from downstream analyses. The combination of the phylogenetic tree and head width data resulted in a phylogeny with 140 tips (113 modern genera + 27 fossil species; figure 2a) that we used for model fitting. We used genus-level information for modern termites because this taxonomic level is representative of the global evolutionary patterns found across termites. We used species-level data for fossils as fossils are scarce, and congeneric species sometimes have different geological time. To account for the influence of within-genus variation on model fitting, we generated 100 sub-datasets by sampling one species for each genus at random. Every evolutionary model was fitted to every 100 sub-dataset.

We used the maximum-likelihood method to fit process-based models of trait evolution on the head width data. We used the models supported by the fitContinuous() function of the R package geiger [39], including unbiased Brownian motion (BM), BM with a directional trend (trend), single optimum Ornstein–Urlenbeck process (OU), lambda, kappa and delta models. Note that we did not use the early burst model, which is also available in the geiger package, because it assumes a similar evolutionary process to delta model and cannot be applied to non-ultrametric trees with more than 100 terminal branches [40]. We also implemented a model that explicitly describes an evolutionary scenario with a unidirectional decrease in body size in lower termites and no directional trend in higher termites (mixed model of trend and Brownian motion: trend-BM). We added this latter model for two reasons. First, higher termites have higher speciation rates [10] and include many

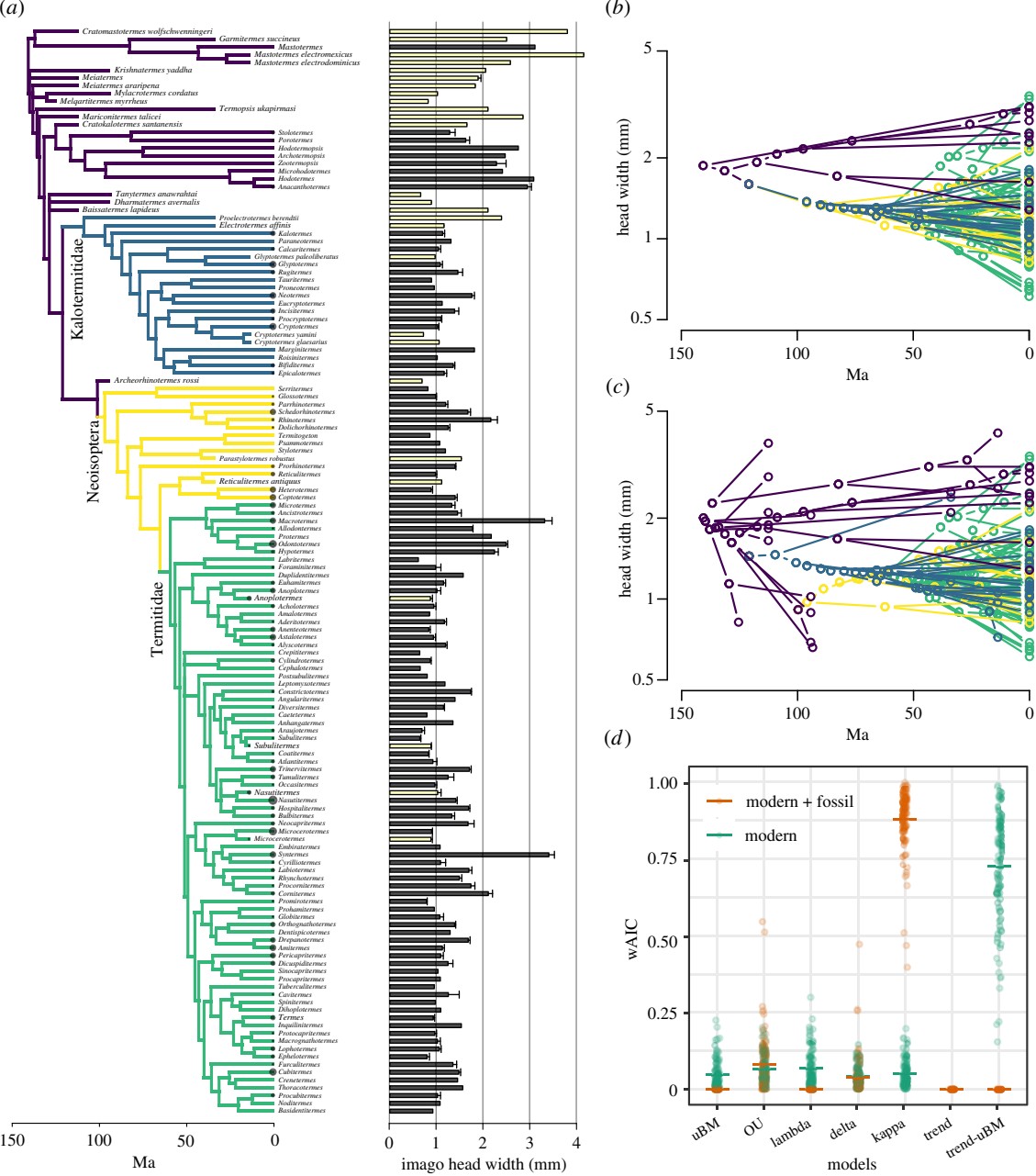

**Figure 2.** Evolution of termite imago head width. (*a*) Phylogenetic tree and head width data used for ancestral state reconstruction and model fitting. The barplots indicate the mean head width (estimated from all species composing each genus) and its associated standard error. Filled bars indicate modern taxa, while open bars indicate fossil taxa. Circles at the tips of the phylogenetic tree represent sample size. (*b,c*) Traitgrams projecting the phylogeny and the evolution of head width in (*b*) modern termite genera and (*c*) modern and fossil termites combined. The traitgrams were generated using the function phenogram() of the R package phytools [38]. (*d*) Akaike weights for seven models fitted on the trees with modern genera only (green) and with both modern genera and fossil species (orange). For model fitting, we generated 100 datasets by subsampling the measurement of one species for each genus at random. The horizontal lines show the mean values. (Online version in colour.)

large species nested within lineages composed of species with small body sizes, which is in direct contradiction with a unidirectional body size reduction model. Therefore, the unidirectional body size reduction model appears to be invalid for higher termites but could be valid for lower termites. Second, a simple unidirectional trend model cannot fit datasets composed exclusively of modern species [41]. We compared the fit of these seven candidate models to (i) the dataset including both modern and fossil species and to (ii) the dataset including only modern species. The support of each model was compared using Akaike weights computed from AICc. We took the natural log values of head width data for model fitting. Note that we also performed these analyses on the head width datasets of workers and soldiers, but only with modern species because fossils of workers and soldiers are excessively rare (electronic supplementary material). All analyses were performed using R v. 4.0.1 [42]. The R scripts

used in this study are available as supplementary material (electronic supplementary material, data S6).

## (d) Statistical analysis

The relationship between head width and social complexity was investigated with Bartlett's test of homogeneity of variance, and phylogenetic generalized least squares (PGLS) using the function pgls() of the R package caper_1.0.1 [43]. Three traits were used as a proxy for social complexity: the presence of true workers, the nesting types, and the colony size. Bartlett's test was used to compare the variance between working castes (workers and pseudergates) and nesting types (one-piece nesters, multiple-piece nesters and central-piece nesters). Bartlett's test is not an optimal method as it does not account for the phylogenetic non-independence among taxa. However, we are not aware of

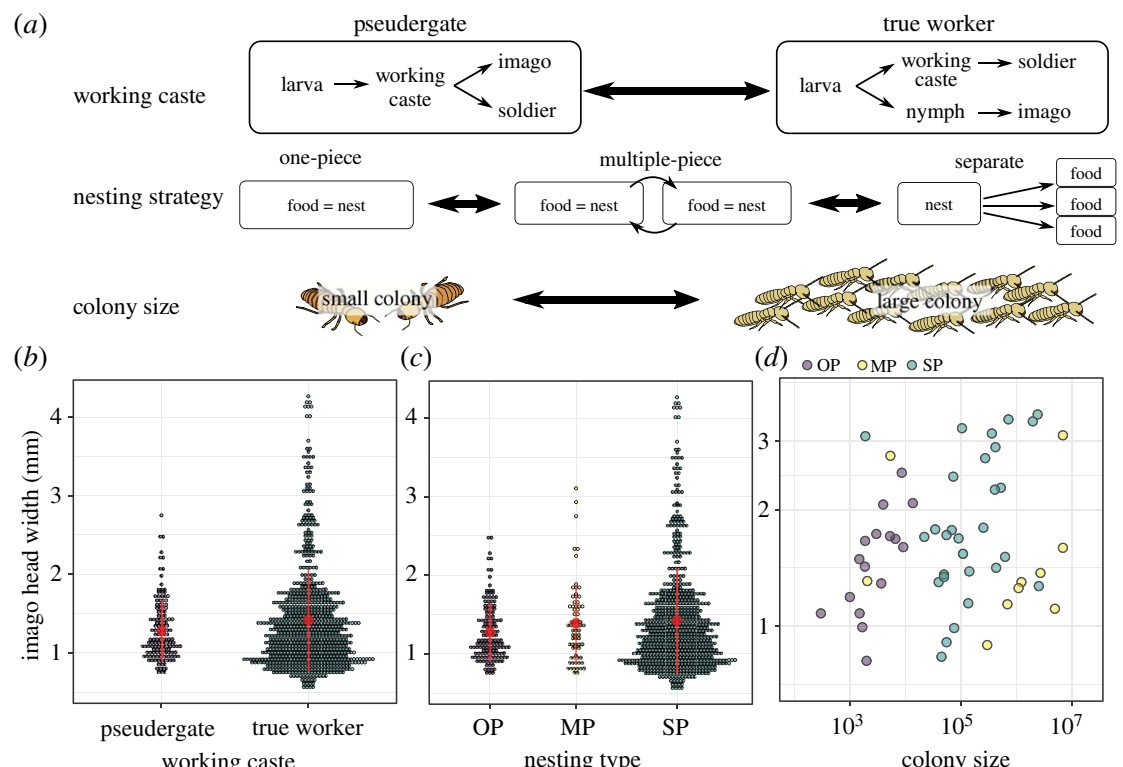

**Figure 3.** Relationship between social evolution and imago body size in termites. (*a*) Factors related to social evolution in termites: working caste, nesting strategy and colony size. Some species lack true workers, and instead have pseudergates that retain the ability to differentiate in alate imagoes. By contrast, true workers are unable to moult into alate imagoes. Three nesting strategies are recognized: one-piece nesters, multiple-piece nesters, and separate-piece nesters. (*b,c*) Comparison of body size across groups with different levels of sociality. Red circles indicate the mean value, and red vertical lines are standard deviations. (*c*) OP, MP and SP indicate one-piece nesters, multiple-piece nesters and separate-piece nesters, respectively. (*d*) Relationship between colony size and imago head width. (Online version in colour.)

equivalent phylogenetic comparative methods. The results of Bartlett's test should be treated with caution. Bartlett's test was performed on the full head width measurement dataset containing species-level information. PGLS was performed on the head width measurement dataset summarized at the genus level and on the genus-level phylogenetic tree described above. We used mean head width values for each genus. We carried out the analyses with Brownian, lambda, kappa and delta models and used the best-fit model identified by AIC. The presence of true workers, nesting types, or colony size was treated as fixed effects. As the social complexity was unknown for fossil species, we only used data obtained from modern species.

We also examined whether size variability among castes was affected by the presence of true workers. We calculated the proportional head width disparity [44] for the pairs imago–worker and imago–soldier using the following equations: (imago head width − worker head width)/worker head width  and  (imago head width − soldier head width)/soldier head width,  respectively. To compare these disparity metrics, we used Bartlett's test and PGLS as described above. We ran PGLS analyses twice, with true workers or pseudergates being the base level, to examine whether level means are significantly different from 0. We also carried out pairwise correlations of head width among castes using PGLS.

## 3. Results and discussion

### (a) Evolution of head width in termites

Using literature data, we compiled a dataset including head width measurements of 1638 termite species. The dataset included 1562 modern termite species (911 imagoes, 1303 soldiers, 840 true workers and 26 pseudergates) and 76 fossil species (69 imagoes, 10 soldiers and two workers). This

dataset comprised nearly half of the described termite species, belonging to 287 genera from all families and subfamilies. The size of this dataset exceeded previous datasets of termite body size by more than one order of magnitude [17,45]. Head width ranged from 0.550 mm to 4.840 mm in imagoes, 0.385 mm to 6.960 mm in soldiers, and 0.459 mm to 3.975 mm in workers (figure 1). The species head width distribution was right-skewed (figure 1), as is the case in many other groups of insects [46].

By fitting various evolutionary models for termite body size evolution, we found that the inclusion of fossil species had large effects on the results. For the dataset of imago head width composed of modern genera only, the best-fit model was a mixed model of trend-BM, which posits a unidirectional body size reduction in lower termites and a non-directional diversification in higher termites (Akaike weight; mean ± s.d. = 0.73 ± 0.19; table 1 and figure 2*b,d*). The trend-BM model was also the best-fit model for the worker head width dataset, while the Ornstein–Uhlenbeck model was the best-fit model for the soldier head width dataset (electronic supplementary material, figures S3 and S4, tables S1 and S2). These results align with the traditional view of the unidirectional decrease in body size hypothesis, at least in imagoes and working castes of lower termites. However, our analyses on the dataset, including both modern and fossil taxa, supported a different scenario for termite body size evolution (figure 2*a,c*). Model fitting showed that the kappa model best explained the imago head width evolution (Akaike weight; mean ± s.d. = 0.88 ± 0.10; table 1 and figure 2*d*). The parameter $\kappa$ was less than 1, indicating that the degree of body size divergence is associated with the number of

**Table 1.** Results of the maximum-likelihood model fitting. The representative parameter values are given for the analyses with and without fossils. For the Ornstein−Uhlenbeck model, the parameter is $\alpha$; for the lambda model, the parameter is $\lambda$; for the delta model, the parameter is $\delta$; for the kappa model, the parameter is $\kappa$. For the trend-uBM model, the parameter is mu, which was estimated from lower termite data. The best-fit model for each dataset is indicated in italics.

| data | model | AICc | wAIC | root | parameter |
|---|---|---|---|---|---|
| modern | Brownian motion | −116.11 (10.13) | 0.047 (0.047) | 1.69 (0.03) | $4.3 \times 10^{-4}$ ($3.8 \times 10^{-5}$) |
| | Ornstein−Uhlenbeck | −117.29 (9.00) | 0.066 (0.047) | 1.58 (0.04) | 0.0067 (0.0017) |
| | lambda | −117.02 (8.78) | 0.068 (0.062) | 1.68 (0.03) | 0.89 (0.054) |
| | delta | −116.32 (9.24) | 0.041 (0.031) | 1.55 (0.04) | 1.9 (0.31) |
| | kappa | −116.62 (9.92) | 0.052 (0.043) | 1.87 (0.06) | 0.64 (0.079) |
| | *trend-uBM* | *−122.58 (8.90)* | *0.73 (0.19)* | *1.84 (0.13)* | $-2.5 \times 10^{-4}$ ($2.1 \times 10^{-4}$) |
| modern + fossil | Brownian motion | −111.43 (7.62) | <0.001 | 1.76 (0.01) | $6.0 \times 10^{-4}$ ($3.2 \times 10^{-5}$) |
| | Ornstein−Uhlenbeck | −122.46 (6.85) | 0.081 (0.09) | 1.73 (0.02) | 0.01 ($7.6 \times 10^{-4}$) |
| | lambda | −112.31 (7.14) | <0.001 | 1.76 (0.01) | 0.97 (0.009) |
| | delta | −120.99 (9.73) | 0.038 (0.061) | 1.97 (0.02) | 0.55 (0.036) |
| | *kappa* | *−128.44 (8.92)* | *0.88 (0.1)* | *2.31 (0.04)* | *0.2 (0.051)* |
| | trend | −112.29 (7.73) | <0.001 | 1.59 (0.02) | 0.0012 ($1.1 \times 10^{-4}$) |
| | trend-uBM | −109.81 (8.34) | <0.001 | 1.61 (0.01) | 0.001 ($6.2 \times 10^{-5}$) |

cladogenetic events [47]. Notably, models representing a unidirectional decrease in body size (trend or trend-BM model) poorly fitted the dataset with fossils included (Akaike weight less than 0.001; table 1 and figure 2*d*). These results reflect the existence of several fossil species with small head widths (e.g. *Melqartitermes*, *Mylacrotermes* and *Tanytermes*) allied to basal termite lineages, contrasting with the modern early-diverging lineage representatives that are large species (figure 2*a,c*). These results highlight the importance of fossil inclusion for an accurate estimation of trait evolution [41].

All models invariably estimated smaller head width for the last common ancestor of modern termites than the head width of the wood roach *Cryptocercus*, the sister group of termites. The kappa model run on the dataset comprising modern and fossil taxa estimated the head width of the last common ancestor of termites at $2.31 \pm 0.04$ mm (table 1). Although 91% of modern termite species are smaller (figure 1), this estimation is half the size of *Cryptocercus* (approx. 5 mm, figure 1). Thus, body size reduction occurred conjointly with the evolution of eusociality in termites [14]. The alternative scenario is that the common ancestor of termites and *Cryptocercus* had a small body size, which implies the subsequent acquisition of larger body size by modern *Cryptocercus*. However, this scenario is unlikely given the comparatively large body size of other wood-feeding cockroach lineages, such as *Salganea* and *Panesthia* [48]. Consequently, the selection pressures acting on small body size were strong at the very origin of termites [17], and weakened since the birth of the common ancestor of modern termites onward.

Our comparative analyses suggested no connection between average body size and traits considered to be linked to advanced sociality in termites. After accounting for the phylogenetic relationship among genera, we found no significant correlation between imago head width and colony size (PGLS, $\lambda$, $\kappa$, $\delta$ transformation, $F_1 = 1.28$, $p = 0.27$; figure 3*d*). Similarly, average imago head width was independent of the presence of a true worker caste (PGLS, $\lambda$, $\kappa$, $\delta$ transformation, $F_1 = 2.884$, $p = 0.09$; figure 3*b*) and of the type of nesting strategy

(PGLS, $\lambda$, $\kappa$, $\delta$ transformation, $F_2 = 2.18$, $p = 0.12$; figure 3*c*). These results were consistent in workers and soldiers (electronic supplementary material, text and figure S3C–E, S4C–E). Thus, caste systems and nesting strategies have no influence on average body size evolution in termites. These results are consistent with the observations that body size is not predictive of social behaviour [49,50]. However, we found that interspecific variation in body size may have increased in termite taxa having traits deemed socially advanced. The interspecific variance of imago head width was significantly higher in termites with a true worker caste (Bartlett's test; $K_1 = 53.88$, $p < 0.001$; figure 3*b*) (variance = 0.42) than in those without true workers (variance = 0.15). Similarly, the interspecific variance of imago head width was significantly different in termites with different nesting strategies (Bartlett's test; $K_2 = 70.66$, $p < 0.001$; figure 3*c*). The variance significantly increased (Bartlett's test; $p < 0.01$ for all pairwise comparisons) along the following sequence: one-piece nesters (variance = 0.13), multiple-piece nesters (variance = 0.24), and separate-piece nesters (variance = 0.44). Note that Bartlett's test does not correct for phylogenetic non-independence among taxa and must therefore be interpreted with caution. Overall, our results suggest that the characteristics of socially complex termite societies do not influence average termite body size but are potentially linked to the emergence of more extreme body size.

## (b) Differentiation of head width among castes

Division of labour among castes is the hallmark of social insects. To determine how social evolution affects termite body size evolution, we compared the head width of castes across termite genera. The head width of imagoes, soldiers, and the working castes strongly correlated to each other across genera (electronic supplementary material, figure S1; PGLS, $p < 0.001$). However, the degree of correlation was dependent on the developmental pathway. Because the developmental lines of alate imagoes and workers diverged early on in the development of genera with true workers,

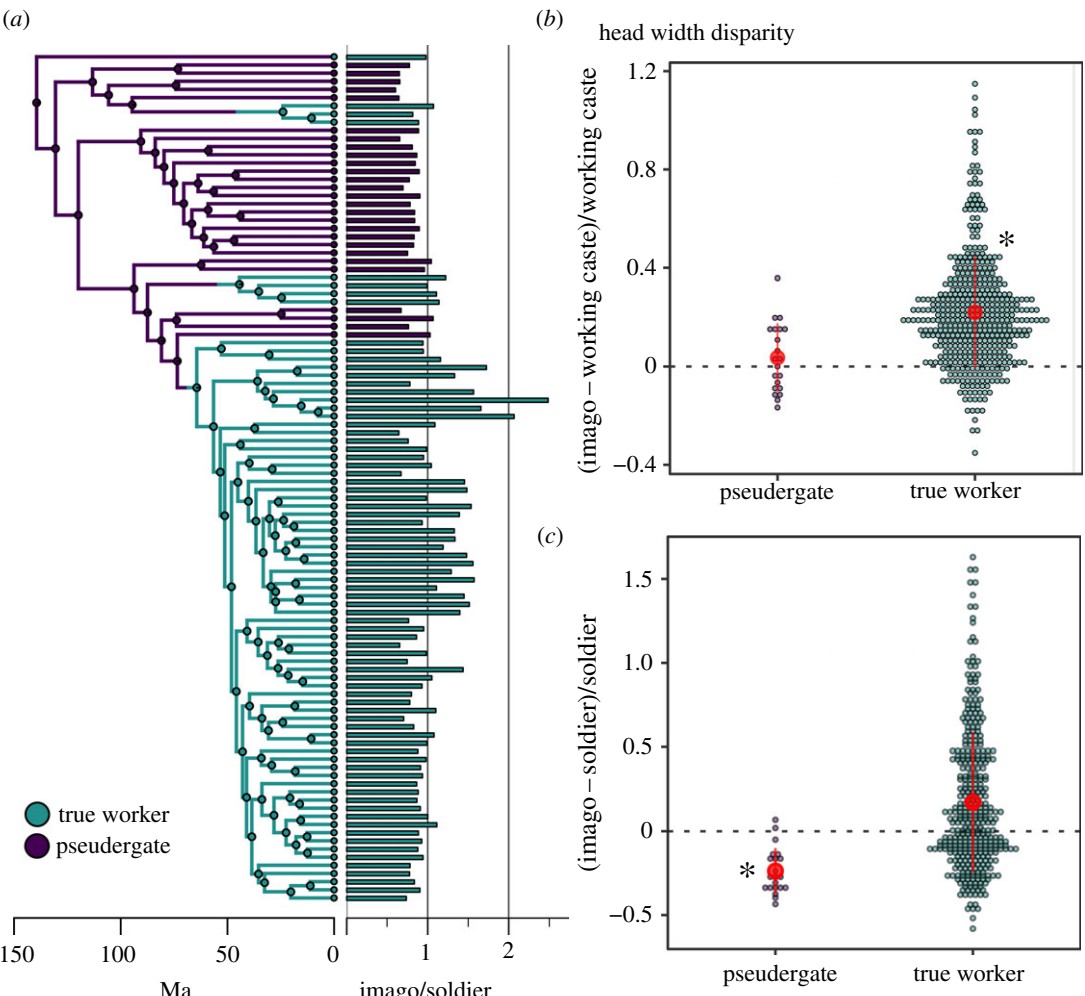

**Figure 4.** Disparity of head width among castes across the phylogenetic tree of termites. (*a*) Ancestral state reconstruction of true workers with head width disparity among castes. The reconstruction was carried out on a genus-level phylogeny, including taxa with data on both imago and soldier head width. Posterior probabilities of ancestral states were estimated using 100 stochastic mappings with the function make.simmap() of the R package phytools [38]. An example of stochastic mappings is shown for the ancestral state reconstruction of each ratio. (*b*) Comparison of imago-working caste head width disparity for genera with pseudergates and true workers. (*c*) Comparison of imago-soldier head width disparity for genera with pseudergates and true workers. Red dots indicate the mean, and red vertical lines indicate the standard deviation. Asterisks denote that the mean of head width disparity is significantly different from 0, which indicates that imago head width is larger than true worker head width, or that soldier head width is larger than imago head width (PGLS; $p < 0.05$). (Online version in colour.)

and because pseudergates can differentiate into alate imagoes through a few moults [25] (figure 3*a*), we expected true workers to have a higher potential for phenotypic diversification. Our comparison between species with pseudergates and true workers revealed that imago head width was larger than pseudergate head width in 52.38% of examined species (11/21), while imago head width was larger than true worker head width in 85.65% of examined species (400/467). In some species with true workers, the imago head width was more than twice that of worker head width (e.g. *Microtermes*) (electronic supplementary material, figure S1). The variance of head width disparity between imagoes and true workers was significantly greater than the variance of head width disparity between imagoes and pseudergates (Bartlett's test, $K_1 = 4.677$, $p = 0.031$; figure 4*b*). After accounting for the phylogenetic relationship among taxa, we found that the mean head width disparity between imagoes and working castes was not significantly different between genera with true workers and genera with pseudergates (PGLS, $\lambda$, $\kappa$, $\delta$ transformation, $F_1 = 3.22$, $p = 0.08$; figure 4*b*), but the mean head width disparity was significantly greater than 0 in genera with true workers (PGLS, intercept, true workers: $t = 2.858$,

$p = 0.005$, pseudergates: $t = 0.797$, $p = 0.428$; figure 4*b*). Therefore, the presence of a true worker caste allows alate imagoes to grow substantially larger than workers and could be associated with a greater variation of head width disparity between alate imagoes and working castes. This link between developmental pathway and size diversity is paralleled in ants [51], suggesting that it represents a common characteristic of body size evolution in social insects.

We carried out the same analyses on imago head width and soldier head width. Because all soldiers differentiate from the working castes, we expected soldiers to have a higher potential for phenotypic diversification in taxa with true workers, as we found was the case for the working castes. We found that most species with pseudergates had larger soldiers than imagoes (92.25%, 119/129), while soldiers were larger than imagoes in fewer species with true workers (46.13%, 280/607). The interspecific variation of head width disparity between imagoes and soldiers was significantly larger in termites with true workers than in termites with pseudergates (Bartlett's test, $K_2 = 14.13$, $p < 0.001$; figure 4*c*). After accounting for the phylogenetic relationship among taxa, we found that the mean head width disparity between imagoes and soldiers was larger in

genera with true workers than in genera with pseudegates (PGLS, $\lambda$, $\kappa$, $\delta$ transformation, $F_2 = 11.32$, $p = 0.001$; figure 4$c$), and that soldiers were on average larger than imagoes in genera with pseudergates, but not in genera with true workers (PGLS, intercept, with pseudergates: $t = -4.790$, $p < 0.001$, with true workers: $t = -0.255$, $p = 0.799$; figure 4$c$). The tendency of soldiers to be larger in species with pseudergates is linked to their one-piece nesting strategies. One-piece nesters are generally defended by phragmotic soldiers that plug galleries with their heads, which, to be efficient, need to have larger heads than other colony members. In contrast, the colonies of multiple-piece nesters and separate-piece nesters extend across larger areas and rely on soldiers employing diverse defensive strategies [52]. The greater variation of head width disparity between alate imagoes and soldiers of species with true workers reflects the diversity of their defensive strategies, each requiring soldiers of different sizes to be optimal.

## 4. Conclusion

Termites have smaller body sizes than other wood-feeding cockroaches. The unidirectional body size reduction hypothesis was believed to be the process behind termite body size evolution, at least in lower termites. However, we found that the unidirectional body size reduction hypothesis is only supported for imagoes and working castes of lower termites when fossil species are excluded from the analyses. Phylogenetic analyses including fossil species indicate that body size evolution was not a unidirectional process. Instead, body size reduction preceded the origin of the last common ancestor of modern termites, which already possessed a smaller body size than cockroaches. Thereafter, the body size of imagoes diversified along with cladogenetic events. Interestingly, a similar pattern was observed for the head width evolution of turtle ants [44]. Our results suggest that the body size range among early termite species was relatively similar to that found in modern termite species.

The acquisition of a diet based on wood, which is uncommon among animals, probably had a modest impact on termite body size diversification, as suggested by the apparent absence of body size reduction and diversification in the wood-feeding *Cryptocercus*, the sister group of termites. In contrast, the evolution of new working castes and nesting types may have been the factors that promoted body size diversification

in termites [9]. Although our analysis did not correct for phylogenetic non-independence among taxa and needs further confirmation from future studies, variation in body size among termite species was greater in taxa possessing a true worker caste and in separate-piece nesters, indicating that social complexity increases body size variation (figure 3).

Body size scales with various traits, including metabolism, abundance and movements [53,54], which can also mediate social interactions between individuals. However, although many models described the interspecific variation of collective building in social insects (in termites, e.g. [55,56]), body size has rarely been implemented as a parameter. Further studies are needed to explore how species with considerable body size differences can build nests of similar size (e.g. *Macrotermes* and *Nasutitermes* build large mounds with the former having head width twice as large as the latter). Also, because closely related species often have similar body sizes, body size is often a confounding variable of other physiological and behavioural traits. To account for the effects of body size, the analysis of large body size dataset alongside phylogeny is essential. Our study paves the way for future comparative studies that aim to shed light on the ecology and evolution of animal society.

Data accessibility. All data are available from the Dryad Digital Repository: https://doi.org/10.5061/dryad.t4b8gtj30 [57]. Electronic supplementary material is on Figshare [58].

Authors' contributions. N.M.: conceptualization, data curation, formal analysis, funding acquisition, investigation, methodology, resources, validation, visualization, writing-original draft; T.B.: conceptualization, data curation, project administration, resources, supervision, writing-review and editing. All authors gave final approval for publication and agreed to be held accountable for the work performed therein.

Competing interests. We declare we have no competing interests.

Funding. This study was supported by a JSPS Research Fellowship for Young Scientists, SPD and CPD (20J00660) to N.M., and OIST core funding.

Acknowledgments. We thank Christine Nalepa for providing head width data; Jigyasa Arora, Ales Bucek, Simon Hellemans, Yukihiro Kinjo and Menglin Wang for help during the data analysis; Ales Bucek for photographs; Tomonari Nozaki and Hiroyuki Shimoji for helpful comments on the data; the members of the Evolutionary Genomics Units at OIST for inspiring scientific discussions; and two anonymous reviewers for constructive comments. N.M. thanks his wife and son for allowing him to manually create a termite head width database during COVID-19 pandemic home quarantine.

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
