## [Peer Review File · Proceedings of the Royal Society B: Biological Sciences]

Review History

RSPB-2021-1458.R0 (Original submission)

Review form: Reviewer 1 (Paul Eggleton)

Recommendation

Accept with minor revision (please list in comments)

Scientific importance: Is the manuscript an original and important contribution to its field?

Good

General interest: Is the paper of sufficient general interest?

Good

Quality of the paper: Is the overall quality of the paper suitable?

Good

Is the length of the paper justified?

Yes

Should the paper be seen by a specialist statistical reviewer?

No

Do you have any concerns about statistical analyses in this paper? If so, please specify them explicitly in your report.

No

It is a condition of publication that authors make their supporting data, code and materials available - either as supplementary material or hosted in an external repository. Please rate, if applicable, the supporting data on the following criteria.

Is it accessible?

Yes

Is it clear?

Yes

Is it adequate?

Yes

Do you have any ethical concerns with this paper?

No

Comments to the Author

See attached file. (See Appendix A)

Review form: Reviewer 2

Recommendation

Major revision is needed (please make suggestions in comments)

Scientific importance: Is the manuscript an original and important contribution to its field?

Good

General interest: Is the paper of sufficient general interest?

Acceptable

Quality of the paper: Is the overall quality of the paper suitable?

Acceptable

Is the length of the paper justified?

Yes

Should the paper be seen by a specialist statistical reviewer?

No

Do you have any concerns about statistical analyses in this paper? If so, please specify them explicitly in your report.

Yes

It is a condition of publication that authors make their supporting data, code and materials available - either as supplementary material or hosted in an external repository. Please rate, if applicable, the supporting data on the following criteria.

Is it accessible?

Yes

Is it clear?

Yes

Is it adequate?

Yes

Do you have any ethical concerns with this paper?

No

Comments to the Author

General comments

The hypothesis being tested here is an interesting one, and it relates to the wider understanding of the evolution of social complexity and of the evolution of superorganismality. The manuscript is, in general, well written, however I am not sure I understand some of the choices made with regards to the analyses. Specifically, I do not think that image head width is necessarily the right measure to understand the evolutionary reduction in body size as sociality grows more complex - it seems to me that the worker would be a more useful focus. I can understand that there do not exist data for workers in the fossil record, however it seems that the authors do have plentiful worker head width data for extant species, and it would be nice to see these data given a similar treatment to the imago head width data (i.e. tested with models describing the unidirectional body size reduction hypothesis). It is not clear to me why the results of models with no phylogenetic component are reported when body size is known to be a trait with a high phylogenetic signal in almost all groups tested.

Line-by-line comments

Lines 36-37 : "...potentially raises the adaptive radiation..." The meaning of this is unclear.

Lines 46-47: Is the unidirectional reduction in body size a hypothesis? Or a known fact? I think that it would be prudent to include the word hypothesis in this sentence, and future references to a unidirectional reduction in body size (i.e. "the unidirectional body size reduction hypothesis") since you state that it "...is believed to be a general evolutionary trend in termites", and indeed appears to be the hypothesis that you are testing here.

Line 49: Perhaps "...allows for improved nutrient economy..." and "...especially in nitrogen..."?

Lines 75-76: Is there any evidence to support the notion that head width correlates with body size? I appreciate the abundance of head width data, but at this point the reader is left wondering if head width is a good proxy for body size as well as being an abundant measurement.

Paragraph starting line 92: Did you take any measures to exclude species based on the size of the series used for taxonomic measurements? I am not particularly familiar with termite taxonomy, but in ant taxonomy type series and reported measurements are sometimes taken from a very small series of specimens (five workers, for example), and so statistical confidence in the mean value may be low.

Lines 171-173: This distinction between the higher and lower termites is interesting, and I think this could, or should, be mentioned in the introduction when defining the hypothesis being tested.

Lines 184-186: It is not completely clear why Bartlett's test was used here. It seems like it was used in order to make use of the species level data for which no phylogeny was available. I have two criticisms of this approach. First, I don't see why a standard least-squares regression wasn't used for consistency with the PGLS model of the genus-level data. Second, body size is well-known to be a trait with high phylogenetic signal, and so a non-phylogenetic analysis of these data is likely to be unreliable in any case.

Lines 254-255: I do not think the content of figure 3 (descriptive plots of the data) supports this statement.

Lines 255-260: I would be very interested to see how these relationships change when the head width of true workers is used in place of imagoes! I suspect that in species with true workers there are divergent selection pressures on worker phenotypes and imago phenotypes and I would not necessarily expect the hypothesised relationships between body size and measures of social complexity to be reflected in imago morphology.

Lines 297-301: To me, this is the result that should be reported. Although the phylogenetically controlled analysis is of a different type to the non-phylogenetically controlled analysis there is clearly a strong phylogenetic signal in head width here, which means that the results of the non-phylogenetic analysis are confounded and thus unreliable. After reading these lines the reader is

left wondering why the results previous (i.e. the non-phylogenetic ones) were reported.
 Lines 301-305: I feel like this fact is a good justification to examine the unidirectional body size reduction hypothesis in workers rather than in imagoes.

Fig 1. The yellow in the key seems to be a different shade of yellow to the yellow used in the histograms.

Fig 2. The branches of the phylogeny and the traitgram are coloured (purple, teal, yellow and green). Although I can infer that the colours are to indicate which parts of the phylogeny are where on the traigram I would expect to see an explanation of the colour scheme for non-termite specialists (presumably this indicates families or some other important clade structure?)

Fig 3. Pseudergate is mis-spelled in this legend.

Decision letter (RSPB-2021-1458.R0)

02-Aug-2021

Dear Dr Mizumoto:

Your manuscript has now been peer reviewed and the reviews have been assessed by an Associate Editor. The reviewers' comments (not including confidential comments to the Editor) and the comments from the Associate Editor are included at the end of this email for your reference. As you will see, the reviewers and the Editors have raised some concerns with your manuscript and we would like to invite you to revise your manuscript to address them.

Research ethics:

Use of animals and field studies:

It is a condition of publication that you make available the data and research materials supporting the results in the article. Please see our Data Sharing Policies (<https://royalsociety.org/journals/authors/author-guidelines/#data>). Datasets should be deposited in an appropriate publicly available repository and details of the associated accession number, link or DOI to the datasets must be included in the Data Accessibility section of the article (<https://royalsociety.org/journals/ethics-policies/data-sharing-mining/>). Reference(s) to datasets should also be included in the reference list of the article with DOIs (where available).

Please submit a copy of your revised paper within three weeks. If we do not hear from you within this time your manuscript will be rejected. If you are unable to meet this deadline please let us know as soon as possible, as we may be able to grant a short extension.

Best wishes,
Professor Loeske Kruuk
<mailto:proceedingsb@royalsociety.org>

Associate Editor

Board Member: 1

Comments to Author:

The two reviewers agree this is an interesting study. However, they raise some important concerns about the analytical methods and the choice of data (i.e. the type of individuals chosen to measure) which I agree need some careful thought and justification.

Reviewer(s)' Comments to Author:

Referee: 1

Comments to the Author(s)

See attached file.

Referee: 2

Comments to the Author(s)

General comments

The hypothesis being tested here is an interesting one, and it relates to the wider understanding of the evolution of social complexity and of the evolution of superorganismality. The manuscript is, in general, well written, however I am not sure I understand some of the choices made with regards to the analyses. Specifically, I do not think that imago head width is necessarily the right measure to understand the evolutionary reduction in body size as sociality grows more complex - it seems to me that the worker would be a more useful focus. I can understand that there do not exist data for workers in the fossil record, however it seems that the authors do have plentiful worker head width data for extant species, and it would be nice to see these data given a similar treatment to the imago head width data (i.e. tested with models describing the unidirectional body size reduction hypothesis). It is not clear to me why the results of models with no phylogenetic component are reported when body size is known to be a trait with a high phylogenetic signal in almost all groups tested.

Line-by-line comments

Lines 36-37 : "...potentially raises the adaptive radiation..." The meaning of this is unclear.

Lines 46-47: Is the unidirectional reduction in body size a hypothesis? Or a known fact? I think that it would be prudent to include the word hypothesis in this sentence, and future references to a unidirectional reduction in body size (i.e. "the unidirectional body size reduction hypothesis") since you state that it "...is believed to be a general evolutionary trend in termites", and indeed appears to be the hypothesis that you are testing here.

Line 49: Perhaps "...allows for improved nutrient economy..." and "...especially in nitrogen..."?

Lines 75-76: Is there any evidence to support the notion that head width correlates with body size? I appreciate the abundance of head width data, but at this point the reader is left wondering if head width is a good proxy for body size as well as being an abundant measurement.

Paragraph starting line 92: Did you take any measures to exclude species based on the size of the series used for taxonomic measurements? I am not particularly familiar with termite taxonomy, but in ant taxonomy type series and reported measurements are sometimes taken from a very small series of specimens (five workers, for example), and so statistical confidence in the mean value may be low.

Lines 171-173: This distinction between the higher and lower termites is interesting, and I think this could, or should, be mentioned in the introduction when defining the hypothesis being tested.

Lines 184-186: It is not completely clear why Bartlett's test was used here. It seems like it was used in order to make use of the species level data for which no phylogeny was available. I have two criticisms of this approach. First, I don't see why a standard least-squares regression wasn't used for consistency with the PGLS model of the genus-level data. Second, body size is well-known to be a trait with high phylogenetic signal, and so a non-phylogenetic analysis of these data is likely to be unreliable in any case.

Lines 254-255: I do not think the content of figure 3 (descriptive plots of the data) supports this statement.

Lines 255-260: I would be very interested to see how these relationships change when the head width of true workers is used in place of imagoes! I suspect that in species with true workers

there are divergent selection pressures on worker phenotypes and imago phenotypes and I would not necessarily expect the hypothesised relationships between body size and measures of social complexity to be reflected in imago morphology.

Lines 297-301: To me, this is the result that should be reported. Although the phylogenetically controlled analysis is of a different type to the non-phylogenetically controlled analysis there is clearly a strong phylogenetic signal in head width here, which means that the results of the non-phylogenetic analysis are confounded and thus unreliable. After reading these lines the reader is left wondering why the results previous (i.e. the non-phylogenetic ones) were reported.

Lines 301-305: I feel like this fact is a good justification to examine the unidirectional body size reduction hypothesis in workers rather than in imagoes.

Fig 1. The yellow in the key seems to be a different shade of yellow to the yellow used in the histograms.

Fig 2. The branches of the phylogeny and the traitgram are coloured (purple, teal, yellow and green). Although I can infer that the colours are to indicate which parts of the phylogeny are where on the traigram I would expect to see an explanation of the colour scheme for non-termite specialists (presumably this indicates families or some other important clade structure?)

Fig 3. Pseudergate is mis-spelled in this legend.

Author's Response to Decision Letter for (RSPB-2021-1458.R0)

See Appendix B.

RSPB-2021-1458.R1 (Revision)

Review form: Reviewer 1

Recommendation

Accept as is

Scientific importance: Is the manuscript an original and important contribution to its field?

Good

General interest: Is the paper of sufficient general interest?

Good

Quality of the paper: Is the overall quality of the paper suitable?

Good

Is the length of the paper justified?

Yes

Should the paper be seen by a specialist statistical reviewer?

No

Do you have any concerns about statistical analyses in this paper? If so, please specify them explicitly in your report.

No

It is a condition of publication that authors make their supporting data, code and materials available - either as supplementary material or hosted in an external repository. Please rate, if applicable, the supporting data on the following criteria.

Is it accessible?

Yes

Is it clear?

Yes

Is it adequate?

Yes

Do you have any ethical concerns with this paper?

No

Comments to the Author

I am very satisfied with the author's responses to my comments on the previous version of the manuscript, and with their additional analyses. I do not have any further comments on the manuscript, and think that it is a good and interesting study.

Decision letter (RSPB-2021-1458.R1)

27-Oct-2021

Dear Dr Mizumoto,

I am pleased to inform you that your manuscript entitled "The evolution of body size in termites" has been accepted for publication in Proceedings B.

Data Accessibility section

Open Access

Paper charges

Yours sincerely,
Professor Loeske Kruuk
Editor, Proceedings B
mailto: proceedingsb@royalsociety.org

Associate Editor:

Board Member: 1

Comments to Author:

The revision reads well and has tackled most of the reviewers' comments adequately.

Board Member: 2

Comments to Author:

(There are no comments.)

Appendix A

Review of termite body size paper - Proceedings

This paper read well. It is clear and to the point, without much wandering from the point. . Although we are not entirely happy with the use of true vs pseudergate workers as factor to examine body size against, I think it's probably the best thing to do in this case and results in some interesting results. However, we feel that this is a pioneering study that will allow more detailed studies in the future. The study would have benefitted from other morphological traits (i.e. a more multidimensional approach) used in the analysis but considering this was purely using the literature it makes sense to use head width (as body size is generally acknowledged as a "super trait", combining in one measure a number of correlated traits) and it clearly provides novel and interesting results, with a number of hypotheses that can be further tested in the future.

We were concerned that the patterns seen here could simply have been generated by differences in speciation rate. More species have been "recently" generated in the Termitidae than other families and maybe that explains the differences between the foragers (true workers) and the wood nesters (pseudergates). Was this "neutral hypothesis" tested for?

I think that the hypothesis relating to true workers could be made a little clearer. We assume that the underlying hypothesis is that the presence of workers, allows the other castes to specialise on other things rather than nest building or foraging. This would be especially true of the soldiers, which would explain the large degree of variation in soldiers. Do any termite species with pseudergates show this sort of variation?

Title

We were a little uncertain about Use of the word "history" in the title – the name implies a chronicle or narrative, which this isn't. It provides data that illuminates history but is not a history in itself.

Introduction

like almost every other termite paper it boils development of termites into two categories, which is likely to be an oversimplification but a common one. Maybe an extra sentence or less to say 'true workers irreversibly deviate from the imaginal line (however some workers are still able to reproduce meaning there is some further complexity to these groupings). No point getting caught up in the contents of Revely et al (2001) paper for this though, as this may complicate a straightforward story.

Methods

Why did you only use the largest morphs when there was polymorphism? This makes a single body size estimate for the soldiers rather hard to rationalise. For example, in the case of *Acanthotermes* there are three soldier morphs – a small one, a medium and a large one. *Hospitalitermes* has three sizes of workers. This choice of the major morphs should be justified or some indication should be given of the error that this approximation contributes to the results.

Queens as the basis for cross species comparisons. Could the authors not have done multiple castes? (compare workers and soldiers across species, for example?)

Results and discussion

Maybe worth having figure 1 in the results section early on, seems a bit out of place in the intro. It may be good to have the first section of figure 3 first in the intro alongside some pictures of termites with different sizes (like in figure 1 but maybe with a few more to let the reader understand the variation involved, maybe even with a *Cryptocercus* photo).

Would be interesting to know not just the interspecific but also the intraspecific variation in body size across castes and subcastes when related to sociality. When the draft refers to interspecific variance does it just mean size of imago heads across a nesting group? Although the paper does go on to talk about this, maybe worth stating more clearly in the introduction what the paper is going to be looking at.

Minor corrections

Ln45. Confusion about numbers of species. Perhaps use “non-termite cockroaches” and “termites”.

Ln95. Why was mid-range value used, rather than the more usual geometric mean?

Ln 315- change ‘in average’ to ‘on average’.

Appendix B

Response to Editor

The two reviewers agree this is an interesting study. However, they raise some important concerns about the analytical methods and the choice of data (i.e. the type of individuals chosen to measure) which I agree need some careful thought and justification.

Reply: We carried out a series of new analyses for this revised version of our manuscript, including new analyses using the data of small morphs (comment 5 of Reviewer 1) and analyses using worker and soldier data (comment 6 of Reviewer 1, comment 1 of Reviewer 2) and body mass data (comment 5 of Reviewer 2).

Response to Reviewer

We found the comments and suggestions from the reviewers to be most helpful and followed all of them while revising the manuscript. We highlighted the parts we modified in the manuscript. The modified parts are referred to below by line numbers.

Response to Reviewer #1:

Comment 1: [...] *We were concerned that the patterns seen here could simply have been generated by differences in speciation rate. More species have been “recently” generated in the Termitidae than other families and maybe that explains the differences between the foragers (true workers) and the wood nesters (pseudergates). Was this “neutral hypothesis” tested for?*

Reply: Thank you for this comment. We recognize the higher speciation rate in the Termitidae. In our analyses, we took into account this effect by explicitly considering the different body size change rates in Termitidae for model fitting (L174-181), or by considering the phylogenetic relationship for true worker and pseudergate comparisons using phylogenetic generalized least squares (L198-200). In Bartlett's test, however, we could not standardise the higher speciation rates in Termitidae, and thus, we mentioned that our analyses are potentially subjected to bias (L193-194, see also the reply to comments 8 and 11 of Reviewer 2).

Comment 2: *I think that the hypothesis relating to true workers could be made a little clearer. We assume that the underlying hypothesis is that the presence of workers, allows the other castes to specialise on other things rather than nest building or foraging. This would be especially true of the soldiers, which would explain the large degree of variation in soldiers. Do any termite species with pseudergates show this sort of variation?*

Reply: Yes, we found that, after being standardised by imago size, soldiers of species with true workers show a larger degree of variation in body size than soldiers of species with pseudergates (Fig. 4C). We rephrased the sentence to explicitly mention this (L312-314).

Comment 3: *Title: We were a little uncertain about Use of the word “history” in the title – the name implies a chronicle or narrative, which this isn't. It provides data that illuminates history but is not a history in itself.*

Reply: We modified the title (L1).

Comment 4: *Introduction: like almost every other termite paper it boils development of termites into two categories, which is likely to be an oversimplification but a common one. Maybe an extra sentence or less to say 'true workers irreversibly deviate from the imaginal line (however some workers are still able to reproduce meaning there is some further complexity to these groupings). No point getting caught up in the contents of Revely et al (2001) paper for this though, as this may complicate a straightforward story.*

Reply: We agree, workers are able to reproduce and often do so, especially in lower termites. We cited this paper and mentioned that true workers are still able to reproduce in some species (L70).

Comment 5: *Methods: Why did you only use the largest morphs when there was polymorphism? This makes a single body size estimate for the soldiers rather hard to rationalise. For example, in the case of Acanthotermes there are three soldier morphs – a small one, a medium and a large one. Hospitalitermes has three sizes of workers. This choice of the major morphs should be justified or some indication should be given of the error that this approximation contributes to the results.*

Reply: As suggested, we added sentences to explain that we used the major morph to keep consistency across species as much as possible (L98). To justify this choice, we clarified that the amount of data with polymorphism was relatively small (12.39%), and using only the smallest subcaste did not change any of our conclusions (L98-100).

Comment 6: *Queens as the basis for cross species comparisons. Could the authors not have done multiple castes? (compare workers and soldiers across species, for example?)*

Reply: Thank you for this comment. In the revised manuscript, we performed analyses on worker and soldier datasets and confirmed that the results on these castes are consistent with that on imago (L184-186, L226-228, SI text, Figs. S3,4, Tables S1,2).

Comment 7: *Results and discussion: Maybe worth having figure 1 in the results section early on, seems a bit out of place in the intro. It may be good to have the first section of figure 3 first in the intro alongside some pictures of termites with different sizes (like in figure 1 but maybe with a few more to let the reader understand the variation involved, maybe even with a Cryptocercus photo).*

Reply: Thank you for this suggestion. We agree with the idea. However, the length of our manuscript almost reaches the limit, and we cannot afford to have an additional figure. We are happy to work on this with further instruction from the editors.

Comment 8: *Would be interesting to know not just the interspecific but also the intraspecific variation in body size across castes and subcastes when related to sociality When the draft refers to interspecific variance does it just mean size of imago heads across a nesting group? Although the paper does go on to talk about this, maybe worth stating more clearly in the introduction what the paper is going to be looking at.*

Reply: We state in the introduction that our research is on body size variation among species (L39-41), and head width is used as a proxy for body size (L75-76). Furthermore, we added analyses on

workers and soldiers in the revised version of this paper (see also reply to the Comment 6).

Comment 9: *Minor corrections: Ln45. Confusion about numbers of species. Perhaps use “non-termite cockroaches” and “termites”.*

Reply: Done (L44).

Comment 10: *Ln95. Why was mid-range value used, rather than the more usual geometric mean?*

Reply: We actually used mean value when it was available from the literature (L92-94). When the mean value was not available, but the range was available, we used mid-range value.

Comment 11: *Ln 315- change ‘in average’ to ‘on average’.*

Reply: Done (L305).

Response to Reviewer #2:

Comment 1: *The hypothesis being tested here is an interesting one, and it relates to the wider understanding of the evolution of social complexity and of the evolution of superorganismality. The manuscript is, in general, well written, however I am not sure I understand some of the choices made with regards to the analyses. Specifically, I do not think that imago head width is necessarily the right measure to understand the evolutionary reduction in body size as sociality grows more complex - it seems to me that the worker would be a more useful focus. I can understand that there do not exist data for workers in the fossil record, however it seems that the authors do have plentiful worker head width data for extant species, and it would be nice to see these data given a similar treatment to the imago head width data (i.e. tested with models describing the unidirectional body size reduction hypothesis). It is not clear to me why the results of models with no phylogenetic component are reported when body size is known to be a trait with a high phylogenetic signal in almost all groups tested.*

Reply: Thank you for the insightful comments. As suggested by reviewer #2, we performed model fitting and subsequent analyses using the worker head width dataset including only modern genera. The conclusion was congruent with the analyses on imagoes, and the unidirectional body size reduction model was supported only in lower termites and without fossil species. We added all these additional analyses in the revised manuscript (L184-186, L226-228, SI text, Figs. S3,4, Tables S1,2). Also, we explained that our motivation for non-phylogenetic Bartlett's tests was to compare the variance between different worker types, and there is no equivalent phylogenetic comparative analysis in our knowledge (L193-197, see also reply to the comment 8 and 11). We mentioned that the results of Bartlett's tests could be subject to potential biased and toned down our arguments (L197, see also reply to the comment 8 and 11). We can remove the Bartlett's tests from the manuscript upon editor's request.

Comment 2: *Lines 36-37 : “...potentially raises the adaptive radiation...” The meaning of this is unclear.*

Reply: We removed this sentence from the revised manuscript (L36).

Comment 3: *Lines 46-47: Is the unidirectional reduction in body size a hypothesis? Or a known fact? I think that it would be prudent to include the word hypothesis in this sentence, and future references to a unidirectional reduction in body size (i.e. “the unidirectional body size reduction hypothesis”) since you state that it “...is believed to be a general evolutionary trend in termites”, and indeed appears to be the hypothesis that you are testing here.*

Reply: We agree with this comment. In the revised manuscript, we use “unidirectional body size reduction hypothesis” throughout the manuscript (L46, 49, 229, and 318).

Comment 4: *Line 49: Perhaps “...allows for improved nutrient economy...” and “...especially in nitrogen...”?*

Reply: We rephrased the part to improve readability (L48).

Comment 5: *Lines 75-76: Is there any evidence to support the notion that head width correlates with body size? I appreciate the abundance of head width data, but at this point the reader is left wondering if head width is a good proxy for body size as well as being an abundant measurement.*

Reply: In the revised manuscript, we added the analysis on the relationship between head width and body mass across 37 termite genera using data published in a previous study (L76-77, SI text, Fig S2). The head width is strongly correlated with body mass, indicating that head width is a good proxy for body size in termites.

Comment 6: *Paragraph starting line 92: Did you take any measures to exclude species based on the size of the series used for taxonomic measurements? I am not particularly familiar with termite taxonomy, but in ant taxonomy type series and reported measurements are sometimes taken from a very small series of specimens (five workers, for example), and so statistical confidence in the mean value may be low.*

Reply: We clarified that we did not exclude any data during literature survey (L94-95). The main focus of this study is to gain insight into the general trend of termite body size evolution by obtaining data from as many species as possible. To this end, we did not focus on estimating the mean head width for every species accurately but on obtaining one representative value for each species, as explained in the methods. Imprecisions in head width measurements for each species is counterbalanced by the large size of analysed datasets and the randomization process we used during analyses.

Comment 7: *Lines 171-173: This distinction between the higher and lower termites is interesting, and I think this could, or should, be mentioned in the introduction when defining the hypothesis being tested.*

Reply: In the revised manuscript, we mentioned that unidirectional body size reduction hypothesis may be applied to lower termites but not to higher termites in the Introduction (L79-81).

Comment 8: *Lines 184-186: It is not completely clear why Bartlett’s test was used here. It seems like it was used in order to make use of the species level data for which no phylogeny was available. I have two criticisms of this approach. First, I don’t see why a standard least-squares regression wasn’t used for consistency with the PGLS model of the genus-level data. Second, body size is well-known to be a trait with high phylogenetic*

signal, and so a non-phylogenetic analysis of these data is likely to be unreliable in any case.

Reply: Thank you for this comment. In the study, we used Bartlett's tests because we want to compare "variance" of head width among termites with different social complexity traits, which cannot be done by standard least-square regression, or by PGLS. We agree with the reviewer's comments and clarified that the result of Bartlett's test is not fully reliable, but we did the best we could (L193-197). We also toned down a few sentences in order to stress the limitation of the test (L197, 261, 268-269, 271, 331-332). We can remove the Bartlett's tests from the manuscript upon request from the editor.

Comment 9: *Lines 254-255: I do not think the content of figure 3 (descriptive plots of the data) supports this statement.*

Reply: As suggested, we did not cite Figure 3 here (L253).

Comment 10: *Lines 255-260: I would be very interested to see how these relationships change when the head width of true workers is used in place of imagoes! I suspect that in species with true workers there are divergent selection pressures on worker phenotypes and imago phenotypes and I would not necessarily expect the hypothesised relationships between body size and measures of social complexity to be reflected in imago morphology.*

Reply: In the revision, we added analyses of the datasets of worker and soldier head width and confirmed that the conclusion is consistent in workers or soldiers (L184-186, L226-228, SI text, Figs. S3,4, Tables S1,2). These results reflect that head width is strongly correlated between castes among species (Fig. S1).

Comment 11: *Lines 297-301: To me, this is the result that should be reported. Although the phylogenetically controlled analysis is of a different type to the non-phylogenetically controlled analysis there is clearly a strong phylogenetic signal in head width here, which means that the results of the non-phylogenetic analysis are confounded and thus unreliable. After reading these lines the reader is left wondering why the results previous (i.e. the non-phylogenetic ones) were reported.*

Reply: Here, we used the former Bartlett tests to compare the variance between worker types, while the later PGLS to compare the mean between worker types. In the revised manuscript, we clearly stated these motivations in the method section (L193-197). Also, we clarify that the results of non-phylogenetic analyses can be subject to bias and needs to be confirmed by future updated measures (L197, 261, 268-269, 271, 331-332, see also the reply to the comment 8).

Comment 12: *Lines 301-305: I feel like this fact is a good justification to examine the unidirectional body size reduction hypothesis in workers rather than in imagoes.*

Reply: In the revised analysis, we fitted a variety of evolutionary models to the worker head width dataset too. We found that the results of the analysis on workers were consistent with those on alate imagoes (L184-186, L226-228, SI text, Figs. S3,4, Tables S1,2).

Comment 13: *Fig 1. The yellow in the key seems to be a different shade of yellow to the yellow used in the*

histograms.

Reply: We confirmed that these two colours have the same RGB values. The different appearance is probably a visual illusion.

Comment 14: *Fig 2. The branches of the phylogeny and the traitgram are coloured (purple, teal, yellow and green). Although I can infer that the colours are to indicate which parts of the phylogeny are where on the traitgram I would expect to see an explanation of the colour scheme for non-termite specialists (presumably this indicates families or some other important clade structure?)*

Reply: Thank you for this helpful comment. We added the notes on the figure (Fig. 2).

Comment 15: *Fig 3. Pseudergate is mis-spelled in this legend.*

Reply: Fixed (L504).